# Third Epidemiological Analysis of Nasopharyngeal Carcinoma in the Central Region of Japan from 2006 to 2015

**DOI:** 10.3390/cancers11081180

**Published:** 2019-08-15

**Authors:** Masafumi Kanno, Norihiko Narita, Yasushi Fujimoto, Naohiro Wakisaka, Tomokazu Yoshizaki, Takeshi Kodaira, Chiyoko Makita, Yuichiro Sato, Keisuke Yamazaki, Takanori Wakaoka, Yuzo Shimode, Hiroyuki Tsuji, Ryosuke Kito, Hajime Ishinaga, Seiji Hosokawa, Hiromasa Takakura, Kunihiro Nishimura, Takuma Matoba, Shigeharu Fujieda

**Affiliations:** 1Department of Otorhinolaryngology Head and Neck Surgery Graduate School of Medical Science University of Fukui, Fukui 910-1193, Japan; 2Otorhinolaryngology/Cognitive and Speech Medicine Nagoya University Graduate School of Medicine, Nagoya University, Aichi 466-8550, Japan; 3Department of Otolaryngology, Head and Neck Surgery Graduate School of Medical Science Kanazawa University, Ishikawa 920-8641, Japan; 4Department of Radiation Oncology Aichi Cancer Center, Aichi 464-8681, Japan; 5Department of Head and Neck Surgery, Niigata Cancer Center Hospital, Niigata 951-8566, Japan; 6Departments of Otolaryngology Head and Neck Surgery, Niigata University Graduate School of Medical and Dental Sciences, Niigata 951-8520, Japan; 7Department of Otolaryngology, Gifu University Graduate School of Medicine, Gifu 501-1194, Japan; 8Department of Head and Neck Surgery, Kanazawa Medical University, Ishikawa 920-0293, Japan; 9Department of Otolaryngology, Shinshu University School of Medicine, Nagano 390-8621, Japan; 10Department of Otorhinolaryngology Head and Neck Surgery, Mie University Graduate School of Medicine, Mie 514-8507, Japan; 11Department of Otorhinolaryngology Head and Neck Surgery, Hamamatsu University School of Medicine, Shizuoka 431-3192, Japan; 12Department of Otorhinolaryngology, Head and Neck Surgery, University of Toyama, Toyama 930-0194, Japan; 13Department of Otorhinolaryngology, Aichi Medical University School of Medicine, Aichi 480-1195, Japan; 14Department of Otorhinolaryngology and Head and neck surgery, Nagoya City University Graduate School of Medical Sciences, Nagoya 467-8602, Japan

**Keywords:** nasopharyngeal carcinoma (NPC), Japan, incidence, survival

## Abstract

The present study aimed to clarify the incidence and clinical outcomes of nasopharyngeal carcinoma (NPC) in the Chubu region of Japan from 2006 to 2015, compared with previous reports. A retrospective analysis was conducted based on medical records from 40 hospitals located in the Chubu region in the central Japanese main island, with a population of around 22.66 million individuals. This study was designed in line with to two previous clinical studies into NPC conducted in the same area of Japan. We recruited NPC patients diagnosed in hospitals across this area over a 10-year period (2006–2015) using a questionnaire about sex, age, primary site, clinical symptoms, pathology, Union for International Cancer Control (UICC) staging, serological exam, treatment, and survival. A total of 620 NPC patients were identified. The age-standardized incidence of NPC from 2006 to 2015 was 0.27 per 100,000 individuals per year. There were no significant differences between this study and the previous two studies conducted in the same area of Japan. The five-year overall survival rate for all patients was 75.9%, while those for patients with stages I, II, III, and IVA were 97%, 91%, 79%, and 68%, respectively. The age-standardized annual incidence of NPC in the present study was 0.27 per 100,000 individuals per year, which was relatively low and stable. The five-year overall survival rate for all NPC patients was significantly improved in this decade compared with previous studies. The smoking rates in male and female NPC patients were 64.5% and 18.8%, respectively, thereby suggesting the involvement of smoking in the incidence of NPC.

## 1. Introduction

Nasopharyngeal carcinoma (NPC), predominantly associated with Epstein-Barr virus (EBV), is characterized by remarkable geographical and racial differences in its incidence. Epidemiological studies conducted over the past several decades have revealed a gradual decline in incidence and a significant reduction in mortality of NPC [1]. However, the increase in population in Asia has led to an increase in the number of deaths due to NPC, from 45,000 in 1990 to 65,000 in 2010 [2]. NPC is more common in some areas of East Asia and Africa [3]. The incidence of NPC is generally less than 1 per 100,000 individuals; however, in southern China it is around 25 per 100,000 individuals, accounting for 18% of all cancers [4]. In Asia, NPC is primarily seen in the middle-aged population, although a high proportion of cases in Africa occur in children. A study of EBV and NPC suggested the existence of a specific interaction between environmental factors such as diet, genetic factors, and viral antibody factors [5]. Pathologically, a vast majority of NPC cases are squamous cell carcinomas (SCCs) showing different degrees of differentiation. They can be classified into three categories based on the World Health Organization (WHO) classification: type 1 is keratinizing SCC; type 2A is differentiated non-keratinizing SCC; type 2B is undifferentiated non-keratinizing SCC, also known as lymphoepithelioma, which is the most common and is most associated with EBV infection; and type 3 is basaloid SCC, which is rare. Patients with all stages of NPC are usually treated using a combination of chemotherapy, radiotherapy, or surgery, including neck dissection of remaining cervical lymph node metastases [6,7]. Most studies have found that chemoradiotherapy leads to better survival than radiotherapy alone. Furthermore, distant metastases and recurrence occur frequently in NPC after treatment [8,9].

NPC is uncommon in Japan; therefore, there have only been a few epidemiological analyses. Sawaki et al. reported that the age-standardized annual incidence of NPC in Japan from 1968 to 1977 was approximately 0.20 per 100,000 individuals per year. Takeshita et al. and Kimura et al. conducted similar studies [10,11], and reported that the age-standardized annual incidence of NPC in the Chubu region of Japan was 0.28 per 100,000 individuals from 1986 to 1995 and 0.29 per 100,000 individuals from 1996 to 2005, respectively. The aim of the present study was to clarify the incidence of NPC in the Chubu area, and to examine the characteristics of NPC in the 10 years from 2006 to 2015 by comparing our data with that of the past two decades.

## 2. Results

### 2.1. Number of NPC Patients

A total of 620 patients were diagnosed with NPC in the 10 years from 2006 to 2015 in the Chubu region. Of these, some were excluded due to insufficient follow-up and lack of records, and a final total of 583 patients were included in the analysis. No significant differences were detected in incidence rate for each year. The patients comprised 474 males and 109 females. Significantly more males than females were diagnosed, with a male: female ratio of 4.3:1 (*p* < 0.05). This sex ratio was greater than the ratio of 2.9:1 previously reported in period from 1996 to 2005 [10]. Figure 1 presents the NPC population according to age and age-specific incidence rates per 100,000 population. The age ranged from 11 to 90 years, and the mean age at diagnosis was 58 years. The incidence rate has been higher in male patients in the age group of 61–70 years.

### 2.2. Age-Standardized Annual Incidence of NPC Per 100,000 Individuals

The age-standardized incidence of NPC for the period 2006–2015 was 0.27 per 100,000 individuals per year. The NPC incidence rates were 0.28 and 0.29 in the previous two studies in 1986–1995 and 1996–2005, respectively [10,11]. There were no significant differences between the three periods.

### 2.3. Patients Histologically Classified according to the 2005 WHO Criteria

The most common pathology was non-keratinizing carcinoma undifferentiated type (WHO type 2B), accounting for 191 cases (35%). There were 172 cases (32%) classified as non-keratinizing differentiated carcinoma (WHO type 2A), 166 cases (31%) classified as keratinizing SCC (WHO type 1), three cases with basaloid SCC (WHO type 3), and six with salivary gland-type carcinomas. There were no cases of nasopharyngeal papillary adenocarcinoma.

### 2.4. Positive Rates of Serum Antibodies to EBV Antigens

We examined the pretreatment serological anti-EBV-VCA IgG titers in 288 of the 583 cases and found 96 cases (33.3%) were positive. The histological proportions of anti-EBV-VCA IgG-positive patients were 48% (49/103 cases) with non-keratinizing undifferentiated carcinoma, 27% (29/106 cases) with non-keratinizing differentiated carcinoma, and 23% (18/79 cases) with keratinizing SCC, resulting in a significant difference *p* < 0.05 (Figure 2). We also examined the pretreatment serological anti-EBV-VCA IgG titer for T classification, UICC stage classification, and anatomical subsites, but found no significant differences (data not shown).

### 2.5. Anatomical Subsites Based on UICC Criteria

We classified the primary tumor site according to UICC anatomical criteria. Primary tumors originated from inferior (0.3%), lateral (47.0%), and posterosuperior (52.0%) sites. There were no significant differences in primary tumor sites compared with those described in the previous two studies [10,11].

### 2.6. Clinical Symptoms

The clinical symptoms of 583 patients were examined and are shown in Figure 3. Otological symptoms were noted in 219 (38%) patients, nasal symptoms in 142 (24%), pharyngeal symptoms in 51 (9%), ophthalmologic symptoms in 33 (6%), cranial neurological symptoms in 36 (6%), headaches in 81 (14%), cervical lymph node swelling in 258 (44%), and other symptoms in 31 (5%), while there were no symptoms in 15 (3%). Ear symptoms included hearing disturbances and a sensation of ear fullness due to otitis media with effusion (OME). Nasal symptoms included nasal obstruction and nasal bleeding. Regarding cranial nerve symptoms, symptoms of abducens nerve palsy was the most obstructive and followed by the trigeminal nerve.

### 2.7. Patient Staging by UICC TNM Classification

UICC staging revealed that stage III was the most common, accounting for 39% (225 cases), followed by stage II (21%; 124 cases). These findings were almost the same as those from the previous studies (Figure 4) [10,11].

### 2.8. Treatments

Cases were divided according to whether they received curative therapy, conservative therapy, or were untreated. We found that 530 cases (90.9%) underwent curative therapy, 34 cases (5.8%) underwent conservative therapy, and 19 cases (3.3%) were untreated. Among the cases treated with conservative therapy, nine cases had insufficient radiotherapy, 13 cases were unable to complete chemotherapy, and others were judged to be insufficiently treated with stage IVB or IVC. Figure 5A summarizes the details of curative therapy. Of the 530 curative cases, only four cases with stage I were not treated with radiotherapy and surgical resection was performed instead. Radiotherapy alone was performed in 39 cases, most of which were stages I and II. There was no case of curative treatment with chemotherapy alone. The breakdown of the chemotherapy used in 487 cases for which curative chemotherapy and radiotherapy were performed is also shown in Figure 5B. The present study did not determine if radiotherapy and chemotherapy were given concurrently or sequentially. In either case, platinum-based regimen was found to be commonly used.

### 2.9. Analysis of Five-year overall Survival Trends for NPC

Figure 6 presents Kaplan–Meier survival curves by tumor stage. The five-year overall survival rate for all patients was 75.9%, and those in stage I, II, and III were 97%, 91%, and 79%, respectively. Some of these survival rates showed a significant improvement of around 10% compared with those in the previous study [10]. The survival rate of stage IV showed an improvement but was still low and was significantly lower compared with other stages (*p* < 0.05).

Analysis by pathology showed there was a significant difference between the five-year overall survival rates of non-keratinizing carcinoma and keratinizing SCC (73.2% vs. 56.0%) (Figure 7).

### 2.10. Relationship between Smoking and Incidence of NPC

The smoking status in 406 of 583 cases was identified. Those with no smoking history and those who have not smoked in the past more than 20 years were considered nonsmokers. Smokers included 207 men and 16 women and nonsmokers included 114 men and 69 women. The smoking rate in male patients was 64.5% and that in female patients was 18.8%. There was no significant difference in overall survival among smokers and nonsmokers (Appendix A). In addition, there were no significant differences in UICC staging and histological classification (Appendix A).

## 3. Discussion

We examined various epidemiological parameters associated with NPC in the Chubu region, where approximately 18% of the total population of Japan reside, and histologically confirmed that 620 patients were diagnosed with NPC in the 10 years from 2006 to 2015. The scale of this Japanese NPC survey is as large as previous reports by Takeshita and Kimura who conducted similar investigations over the past two decades [10,11]. Therefore, the present study demonstrates the results of a 30-year follow-up in the same area. The age-standardized annual incidence of NPC in the present study was 0.27 per 100,000 individuals per year, whereas it was 0.28 in Takeshita’s report during 1986–1995 [10] and 0.29 in Kimura’s report during 1996–2005 [11], exhibiting just a slight change over 30 years. The mean age at diagnosis of NPC in the present study was 58.0 years, whereas it was 54.1 and 55.2 years in the previous two reports [10,11]. The average age tended to increase slightly, probably due to the increase in average life expectancy in Japan. Although the sex ratio in the present study was 4.3:1, indicating that the present study included more males than previously reported (2.9:1) [10], there was no significant difference according to the chi-square test. In the previous two studies, the ratio of males to females was 2.8–2.9:1 [10,11], and was reported to be 2.6:1 in Hong Kong 2006 [5] and 2.1–2.7:1 in south China 2011 [12], which is considered a high NPC risk area. However, according to the latest NPC epidemiological survey in China, the sex ratio is reported as 2–5:1 [13], and our results are similar. Although the reason why this change in sex ratio has significantly changed in this decade cannot be clearly stated, the involvement of oral and oropharyngeal cancer has been considered in Japan. According to the Cancer Registry and Statistics, Cancer Information Service, National Cancer Center, Japan, the male: female ratio for oral and oropharyngeal cancer has increased from 2.09 in 2009 to 2.62 in 2012 (https://ganjoho.jp/public/index.html). Attention must be paid to future transition in terms of whether it will be biased to male dominance as it is. The 2005 version of the WHO pathology classification is the most recent version, and previous WHO classifications, such as type I (SCC), type II (non-keratinizing carcinoma), or type III (undifferentiated carcinoma) are no longer used. In general, WHO I is equivalent to the current group of keratinizing SCC (type 1), WHO II is non-keratinizing carcinoma differentiated type (type 2A), and WHO III is non-keratinizing carcinoma undifferentiated type (type 2B). In Kimura’s report, 36% were classified as WHO I (current type 1), 27% were WHO II (type 2A), and 37% were WHO III (type 2B). Comparing this study with previous studies reveals only a slight change in the percentage of histological diagnoses [10,11]. Keratinizing SCC is more common in the lower incidence areas compared with the high incidence areas and has been reported to be >25% [14,15,16]. In contrast, keratinizing SCC is in the minority in the high incidence areas, and reported to be <10% [17,18]. In the present study, there was a 31% incidence of keratinizing SCC (WHO type 1) in the Chubu region of Japan. Keratinizing SCCs are predominant in low-risk populations and are thought to occur primarily due to environmental exposure such as smoking as well as other head and neck cancers, and the presence of EBV infection is also indicated as part of the incidence factor [19,20]. NPCs in young adults are predominantly non-keratinizing carcinomas and have familial predisposition. EBV infection is involved in the stages following either germline mutations or minor genetic polymorphic variants [1,21,22]. Undifferentiated carcinoma, common to high-risk populations, includes early genetic polymorphisms associated with chronic (dietary) exposure to nitrosamine compounds [23,24]. EBV infection is also considered to be an important step in the pathway of undifferentiated malignant tumors [3,25]. Serological anti-EBV-VCA IgG titers in the present study are presented in Figure 2. In total, 18 of 79 (23%) patients with keratinizing SCC were positive, whereas 78 of 209 (37%) patients with non-keratinizing carcinoma patients were positive, indicating a significant difference (*p* < 0.05). Furthermore, dividing non-keratinizing carcinoma into non-keratinizing differentiated carcinoma (WHO type 2A) and undifferentiated carcinoma (WHO type 2B) led to the positive rate becoming noticeable. Undifferentiated carcinoma revealed that about half of the cases showed anti-EBV-VCA IgG titers >640. This result was particularly greater than non-keratinizing differentiated carcinoma and keratinizing SCC (*p* < 0.01). The previous reports by Kimura and Takeshita also showed similar significant differences [10,11].

The primary NPC site was classified into three parts, and in line with the previous studies, demonstrated that lateral and posterosuperior types were the major sites. It is crucial to diagnose the origin site of NPC, because the appearances of clinical symptoms are different depending on the site of origin. Our results were almost the same as those found in the previous two reports; the most frequent complaint was cervical lymph node swelling followed by ear and nasal symptoms. Otological signs included a sensation of fullness of the ear due to OME. Therefore, it is necessary to examine the nasopharynx carefully when treating middle- and old-aged patients with recurring OME. However, it is difficult to diagnose early stages because subjective symptoms do not appear until a tumor grows to certain size and it is not feasible to examine the nasopharynx in all patients with a common cold. In the present study, advanced stage III and IV cancers accounted for 70%, similar to the 69% found in the previous report by Kimura [10]. In Takeshita’s report, the proportion of advanced cancer was even higher, accounting for 87% [11]. Considering that the UICC stage classification of NPC has changed very little over the last 20 years, this suggests that the development and use of the endoscope has enabled early checkups by doctors. In Japan, the use of positron emission-tomography/computed-tomography (PET/CT) as part of medical checkup is not widely spread even now, and its use as an insurance medical treatment is limited after the pathological diagnosis is confirmed, thereby indicating that PET/CT does not contribute to early diagnosis.

Radiotherapy remains the mainstay of treatment, and the addition of chemotherapy to radiotherapy has significantly improved overall survival [26,27,28,29]. As Figure 6 shows, the five-year overall survival improved from 67.6% to 75.9% from previous studies. We believe this is due to advances in radiotherapy and treatment of the side effects of chemotherapy. In all 40 hospitals, the standard treatment of NPC was a combination of radiotherapy and chemotherapy. In the present study, the treatment completion rate of radiotherapy and chemotherapy was very good, as there were only nine cases of incomplete radiotherapy and 13 cases of incomplete chemotherapy. In Japan, intensity-modulated radiation therapy (IMRT) was shown to be more common for head and neck cancer since 2005. PET/CT and magnetic-resonance-imaging diagnosis the staging correctly, and IMRT reflects it in the treatment effect. Currently, most cancer treatment hospitals use the IMRT system, apparently resulting in improved tumor control and reduced toxic effects [30]. Meanwhile, platinum-based chemotherapy was selected at all 40 centers. The addition of platinum-based chemotherapy improves disease management but is associated with significant early toxic effects such as mucosal damage, dysphagia, nausea, fatigue, immune depression and fever, that cause interruptions in treatment. We believe the development of support therapy for the side effects of chemoradiation, and team support for nutrition, oral care, pain treatment, and mental care have increased the completion rate of platinum-based chemoradiotherapy. The five-year survival rates of stage IV cancers were significantly lower compared with other stages. Patients with early NPC stages showed significantly improved survival compared with patients diagnosed at advanced disease stages. Furthermore, the primary tumor capacity was closely related to the survival rate of NPC patients. Early diagnosis of NPC should improve the cure rate and reduce morbidity and metastasis [31,32,33]. This study has certain limitations. First, the execution rate is uncertain even though IMRT is implemented at most facilities. Although chemotherapy and radiotherapy have concurrent treatment at most centers, the timing of the combination is ambiguous and the amount of medicine used is inconsistent, which can be another potential bias.

The association between smoking and NPC has been previously reported [34]. The smoking rates in Japan are annually published by Japan Tobacco Inc. (JT). Smoking rates have gradually declined, with the rate in Japanese smokers decreasing from 41.3% to 31.0% for men and from 12.4% to 9.6% for women from 2006 to 2015. The smoking rate in this study was high at 64.5% for men and 18.8% for women, thus suggesting the involvement of smoking in the incidence of NPC.

Epidemiological studies conducted over the past decades reveal a gradual decline in the incidence of NPC and a significant reduction in mortality [35]. Advances in radical treatment (particularly radiation therapy and chemotherapy) have also had a major impact on improving clinical outcome, resulting in long-term survival of patients with nasopharyngeal carcinoma. However, treatment and management methods are still being developed, and we hope that advancements in research will provide clear conclusions. Epidemiological research on rare cancers is very important. We believe that this research provides epidemiological clinical data of NPC in Japan and can be the basis of various research studies. We also hope that the findings of our study will help promote the control of rare cancers in Japan.

## 4. Materials and Methods

We surveyed the medical records of the 40 hospitals in Chubu region, including nine prefectures (Niigata, Toyama, Ishikawa, Fukui, Nagano, Gifu, Shizuoka, Aichi, and Mie). The selected hospitals were the cancer treatment centers recommended by the Japanese Ministry of Health, Labor and Welfare. All cases were histologically proven nasopharyngeal primary malignant epithelial tumors diagnosed for the first time at each hospital between 2006 and 2015. Cases diagnosed as reappearance of a previously treated tumor were excluded. The average population of the Chubu region was 22.66 million based on the 2005, 2010, and 2015 national censuses. The average populations of each of the prefectures were 2,390,000 in Niigata, 1,090,000 in Toyama, 1,170,000 in Ishikawa, 810,000 in Fukui, 2,150,000 in Nagano, 2,070,000 in Gifu, 3,750,000 in Shizuoka, 7,380,000 in Aichi, and 1,850,000 in Mie. In line with previous studies [10,11], our study measured variables including sex, age, site, WHO histological criteria, positive rates of serum antibodies to EBV-related virus capsid antigen (EBV–VCA), Union for International Cancer Control (UICC) TNM staging, clinical symptoms, and five-year survival rate. In addition, our study additionally examined the association of incidence of NPC with smoking. According to the Japanese lung cancer survey, the risk of onset is equivalent to that in nonsmokers who have quit smoking for >20 years [36], and in this survey, individuals with no smoking history in the past >20 years were also considered as nonsmokers. Positive rates of serum anti-EBV-VCA immunoglobulin (Ig) G titers were defined as ≥640 mg/dL. TNM classification and anatomical subsites were based on the UICC criteria, 7th edition. Statistical analysis: All cases were analyzed using Stat View according to sex, age, site, histological type, EBV positive rate, stage, and clinical symptoms. The level of significance between various parameters was assessed using chi-squared test to obtain *p*-values. Significant difference was indicated by *p*-values < 0.05. The Kaplan–Meier method and log rank test were used to analyze survival curves. This study was approved by University of Fukui Clinical Research Review Board (No. 20160105). In addition, we also received approval from the Ethic Committee of each facility.

## 5. Conclusions

The age-standardized annual incidence of NPC in the present study was 0.27 per 100,000 individuals per year, which was relatively low and stable. The five-year overall survival rate in all NPC patients was significantly improved in this decade compared with that reported in previous studies.

## Figures and Tables

**Figure 1 cancers-11-01180-f001:**
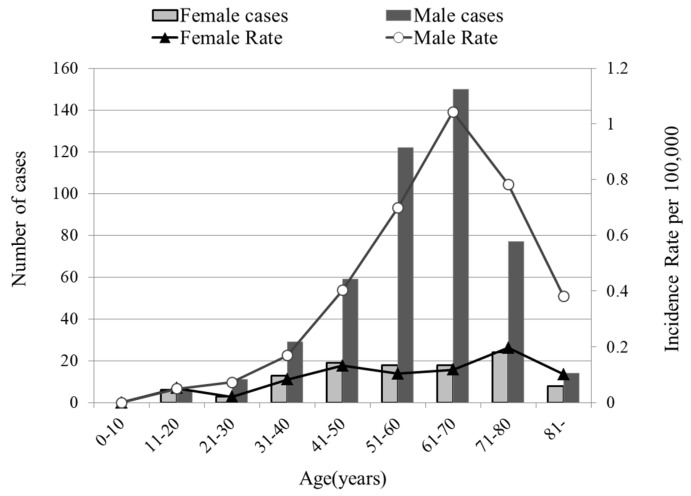
Number of new cases per age group according to age-specific incidence rates per 100,000 population.

**Figure 2 cancers-11-01180-f002:**
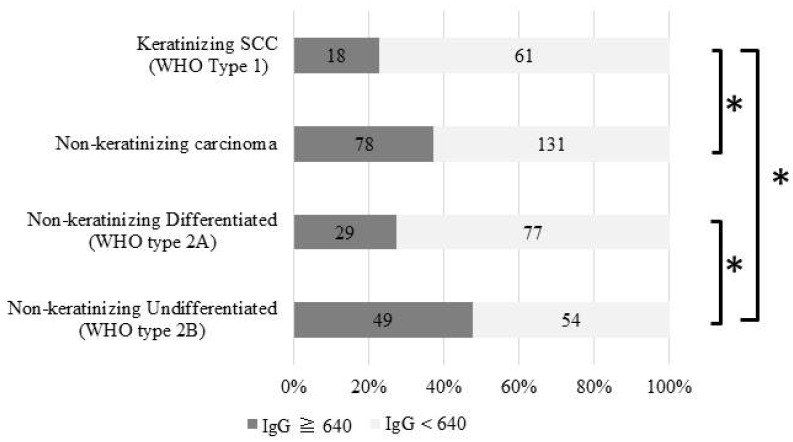
Positive rates of serum antibodies to Epstein–Barr-related antigens. Gray boxes represent anti-EBV-VCA IgG titers of ≥640 mg/dL, and white boxes represent titers of <640 mg/dL. Numbers in the boxes indicate the number of cases. * Fisher test: *p* < 0.05.

**Figure 3 cancers-11-01180-f003:**
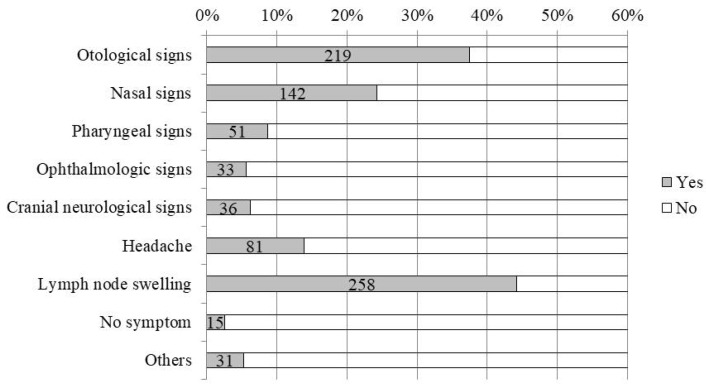
Clinical symptoms. The proportion of each symptom is shown on the horizontal axis. The maximum value on the horizontal axis was adjusted to 60%. Gray boxes indicate patients with symptoms, and the numbers indicate the number of cases.

**Figure 4 cancers-11-01180-f004:**
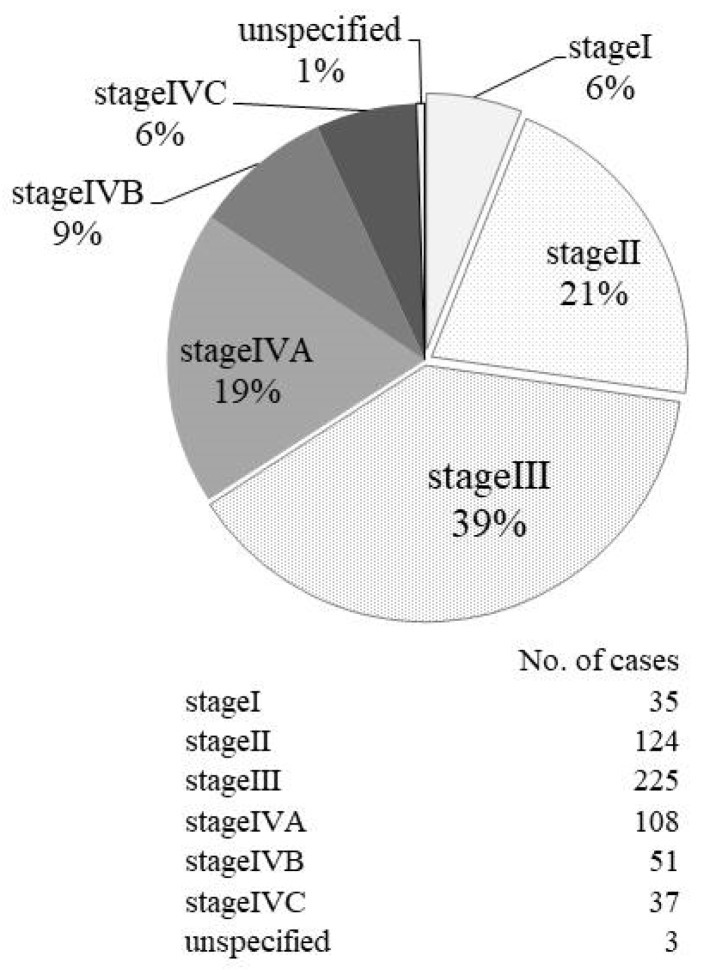
Pretherapy clinical staging. Patient staging distribution according to the TNM classification of UICC 7th Edition.

**Figure 5 cancers-11-01180-f005:**
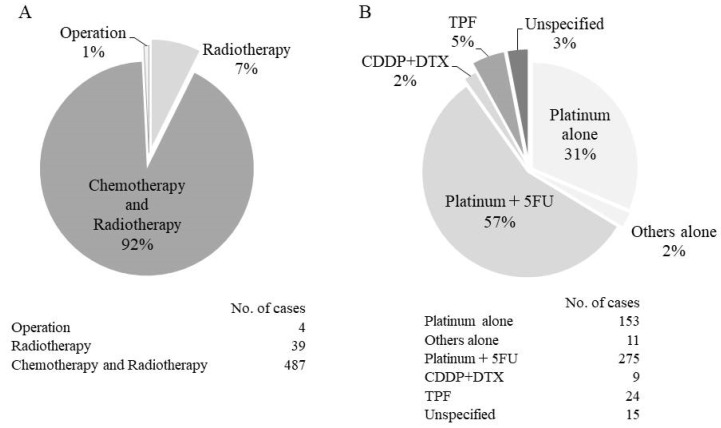
Details of curative treatments: (**A**) The details of curative therapy; (**B**) the breakdown of the chemotherapy used in 487 cases in which curative chemotherapy and radiotherapy were performed. Platinum includes cisplatin, carboplatin, and nedaplatin. Others alone includes eight cases of S-1 and three cases of DTX. CDDP, cisplatin; DTX, docetaxel; 5FU, 5-fluorouracil; S-1, tegafur/gimeracil/oteracil; TPF, docetaxel/cisplatin/fluorouracil.

**Figure 6 cancers-11-01180-f006:**
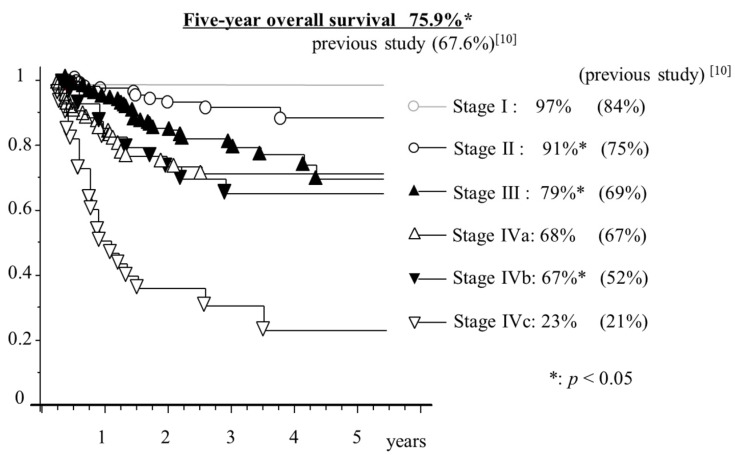
Kaplan–Meier curve for overall survival according to stages. Asterisks indicate significant improvement in our study compared with that reported in a previous study (*p* < 0.05).

**Figure 7 cancers-11-01180-f007:**
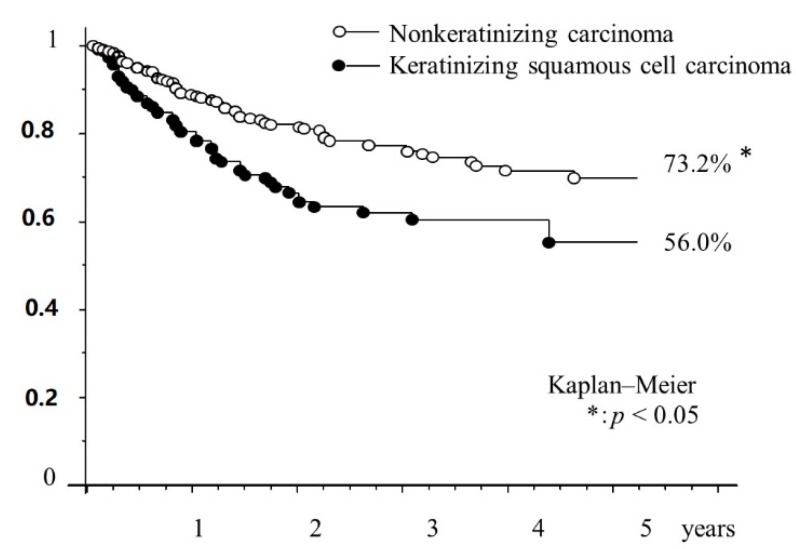
Kaplan–Meier curve for overall survival according to pathology. Asterisks indicate significant difference compared with the pathology (*p* < 0.05).

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
