# Peer review of "Third Epidemiological Analysis of Nasopharyngeal Carcinoma in the Central Region of Japan from 2006 to 2015"

_cancers, 2019, doi:10.3390/cancers11081180_

Round 1

Reviewer 1 Report

The authors present data from 10 years retrospective epidemiological surveillance of nasopharyngeal carcinoma in a Japanese region accounting for 22 million inhabitants.

The study seems methodologically well done and reporting quality is fine.

Some minor comments:

In the discussion section could be useful to highlight some limitations of the study. Also, it would be interesting a brief discussion of the impact of such analysis form a clinical and a public health perspective.

Author Response

We would like to thank Reviewer 1 who identified areas of the manuscript that needed corrections or modification. We hope that our revisions along with our responses address your concerns and that our revised manuscript is now suitable for publication.

A. 

Based on your suggestion, we have added more comments about study limitations in lines 310–313.

Wehavealso commented about the studycontributions to clinical and public health inlines 326–329.

Reviewer 2 Report

I apprecite to review the manuscript. I have 3 questions.

1) Smoking is a important risk factor of NPC. The authors had better to incorporate smoking into the analysis

2) The authors insisted that advances in radiotherapy and treatment of the side effects of chemotherapy was the cause of survival improvement of stage I,II, III NPC. But stage shifting after itoduction of more accurate diagnostic tool such as PET scan is well known survival data improvement factor. So, the authors had better to present data about staging diagnostic methods compared with previous decades epidemiologic reports.

3) High male patients ratio (4.3 : 1)!. Can you comment why male patients fraction increased surprisingly?

Author Response

Reviewer #2: 

We would like to thank Reviewer 2 who identified areas of the manuscript that needed corrections or modification. We hope that our revisions along with our responses address your concerns and that our revised manuscript is now suitable for publication.

Q1.

 Smoking is an important risk factor of NPC. The authors had better to incorporate smoking into the analysis

 A1. 

Based on your suggestion, we re-investigatedsmoking history for almost all thehospitals.As a result, the relationship of incidence of NPC with smoking was clarified in approximately406 of the583 cases analyzed. The results are presentedin lines 203–210. Along with this, we havealso added sentence in the Abstract in lines55 and 56. As you pointedout, the relationship between smoking and incidence of NPC has been reported. The same result was obtained in this re-examination. However, there was no significant difference in survival rate, histopathology,and staging. We have included this informationin lines 314–319 of the Discussion section.The definition of smoking has beenadded to the Materials and Methods sectionin lines 344–347.

 Q2. 

The authors insisted that advances in radiotherapy and treatment of the side effects of chemotherapy was the cause of survival improvement of stage I,II, III NPC. But stage shifting after itoduction of more accurate diagnostic tool such as PET scan is well known survival data improvement factor. So, the authors had better to present data about staging diagnostic methods compared with previous decades epidemiologic reports.

A2. 

In Japan, the use of PET testing is strictly limited, and basically,it has to be performedafter having a pathological diagnosis; therefore,PET does not contribute to early diagnosis.

The use of PET in self-consulting medical care(that is, medical checkup)is not widespread; pleaserefer the graph presentedbelow. This information published by Japan RI Association wastranslated into English. Moreover, there is no evidencethat the diagnosis has come earlier than in the previous survey. We can diagnose early compared with the survey 20 years ago, but this is just the time of development of endoscope in Japan, and we comment that way. This information has been included inlines 277–285.

 In addition, PET's contribution to the accuracy of diagnosis is already mentioned inlines 294–296.

 Q3. 

High male patients ratio (4.3 : 1)!. Can you comment why male patients fraction increased surprisingly?

 A3. 

Certainly,the incidence rate in male patientsis high. A recent Chinese survey published in Lancet has also reported asignificantly high incidence rate inmale patientsthan infemalepatients. Moreover, in Japan, it appearsthat the male ratio is increasing with regard to incidence rate of all oropharyngeal cancers. This information was mentioned inlines 226–235.

Reviewer 3 Report

This is an updated epidemiological analysis of nasopharyngeal carcinoma in the central region of Japan from 2006 to 2015. The materials and methods are appropriate and the results are clearly presented. The results of this study can be used for comparative study in different regions. It is acceptable for publication.

Author Response

Reviewer #3: 

We would like to thank Reviewer 3 who identified areas of the manuscript that needed corrections or modification. We hope that our revisions along with our responses address your concerns and that our revised manuscript is now suitable for publication.

This is an updated epidemiological analysis of nasopharyngeal carcinoma in the central region of Japan from 2006 to 2015. The materials and methods are appropriate and the results are clearly presented. The results of this study can be used for comparative study in different regions. It is acceptable for publication.

A.

We are very grateful toyou for reviewingour manuscriptandforyour polite and valuable comments.

Round 2

Reviewer 2 Report

Thank you for your sincere and adequate revision.